# Inhibitors of Farnesyl Diphosphate Synthase and Squalene Synthase: Potential Source for Anti-*Trypanosomatidae* Drug Discovery

**Boniface Pone Kamdem** [1,2,*] and **Fabrice Fekam Boyom** [1]

1 Laboratory for Phytobiochemistry and Medicinal Plants Studies, Department of Biochemistry, Faculty of Science, University of Yaoundé 1, Yaoundé P.O. Box 812, Cameroon; fabrice.boyom@fulbrightmail.org
2 Department of Basic and Applied Fundamental Sciences, Higher Institute of Agriculture, Forestry, Water and Environment (HIAFWE), University of Ebolowa, Ebolowa P.O. Box 755, Cameroon
* Correspondence: ponekamdemboniface@gmail.com; Tel.: +237-680-98-76-69 or +237-657-52-57-18

**Abstract:** Trypanosomatids are mainly responsible for leishmaniasis, sleeping sickness, and Chagas disease, which are the most challenging among the neglected tropical diseases due to the problem of drug resistance. Although problems of target deconvolution and polypharmacology are encountered, a target-based approach is a rational method for screening drug candidates targeting a biomolecule that causes infections. The present study aims to summarize the latest information regarding potential inhibitors of squalene synthase and farnesyl phosphate synthase with anti-*Trypanosomatidae* activity. The information was obtained by referencing textbooks and major scientific databases from their inception until April 2023. Based on in vitro experiments, more than seventy compounds were reported to inhibit squalene synthase and farnesyl diphosphate synthase. Among these compounds, more than 30 were found to be active in vitro against *Trypanosomatidae*, inferring that these compounds can be used as scaffolds to develop new drugs against trypanosomatid-related infections. Overall, natural and synthetic products can inhibit enzymes that are crucial for the survival and virulence of trypanosomatids. Moreover, in vitro experiments have confirmed the activity of more than half of these inhibitors using cell-based assays. Nevertheless, additional studies on the cytotoxicity, pharmacokinetics, and lead optimization of potent anti-Trypanosomatid compounds should be investigated.

**Keywords:** Trypanosomatids; squalene synthase; farnesyl diphosphate synthase; target-based approach; neglected tropical diseases





## 1. Introduction

Neglected tropical diseases (NTDs) are a collection of 20 diverse conditions that are prevalent in subtropical and tropical regions, where they affect more than 1 billion people who live in impoverished communities. Most of these diseases are parasitic, with high endemicity in developing countries in Africa, Asia, and the Americas [1]. Leishmaniasis and trypanosomiases are among the NTDs that occur in several sub-Saharan African countries [2]. These diseases are caused by trypanosomatids that interact with several mammals and insects to complete their life cycles. In general, *Trypanosoma brucei* gambiense and *Trypanosoma brucei* rhodesiense, *Trypanosoma cruzi*, and *Leishmania* spp. are the parasites causing human African trypanosomiasis (HAT or sleeping sickness), Chagas disease, and cutaneous, mucocutaneous, and visceral leishmaniasis, respectively. These kinetoplastid ailments are maladies of poverty that have received limited funding for the discovery and development of new tools [3]. The number of people infected by *T. cruzi* is estimated between 6 and 7 million, and the disease is mostly transmitted by contact with the feces and/or urine of infected blood-sucking triatomine bugs (vector-borne transmission) [4]. The number of new cases of HAT has been reduced to the lowest level (<1000 new cases)

in 2020, with 60–70 million people at risk of infection in thirty-six African countries. This disease is transmitted by infected tsetse flies [5]. Regarding leishmaniasis, transmission occurs through bites by infected female phlebotomine sandflies, leading to estimates of approximately 700,000 to 1 million new cases annually [6]. Current treatments for human African trypanosomiasis include pentamidine, suramin, melarsoprol, eflornithine, and nifurtimox, whereas benznidazole and nifurtimox (Figure 1) are indicated for Chagas disease's treatment [4,5]. Approved therapies for leishmaniasis's treatment include intravenous amphotericin B (liposomal) for VL and miltefosine (oral) for cutaneous leishmaniaisis (CL), mucocutaneous leishmaniasis (ML), and visceral leishmaniasis (VL) caused by specific species [7]. However, these medications suffer from the limitations of toxicity, requirements for parenteral administration, variable efficacy, and length of treatment schedules, in addition to the problem of drug resistance [2]. Therefore, there is a pressing need to search for safer anti-*trypanosomatidae* therapies. Numerous studies have recently demonstrated the effectiveness of natural products vis-à-vis *trypanosomatidae* at preclinical and clinical levels. Some of these studies focused on the identification of enzymes that are crucial for the survival and virulence of *trypanosomatidae*. These enzymes include farnesyl diphosphate synthase and squalene synthase [8].

**Figure 1.** Chemical structures of commonly used anti-*Trypanosomatidae* drugs.

In previous studies, 2-alkyl aminoethyl-1,1-bisphosphonic acids [9] and quinuclidine-based compounds [10,11] were reported to inhibit squalene synthase, whereas taxodione

and arenarone inhibited the enzyme farnesyl diphosphate synthase [12]. Unlike humans, these parasites (*T. cruzi*, *T. brucei*, and *Leishmania* spp.) use various types of sterols (e.g., ergosterol, 24-ethyl-cholesta-5,7,22-trien-3β-ol and its 22-dihydro analogs) in lieu of cholesterol in their cell membranes, so inhibiting the biosynthesis of endogenous sterol might be an essential therapeutic target [13].

The identification of inhibitors of these crucial enzymes might afford potentially active anti-*Trypanosomatidae* hit compounds that can serve as starting points for the discovery of new treatments against leishmaniases and trypanosomiases. Thus, reviews or monographs are needed to summarize and discuss the latest information regarding the possibility for inhibitors of farnesyl diphosphate synthase and squalene synthase to impede the growth of *Trypanosoma* and *Leishmania* species.

To date, only a few reports [13,14] have previously been published on this topic. Thus, this work aims to recapture up-to-date information regarding potential inhibitors of squalene synthase and farnesyl phosphate synthase with anti-*Trypanosomatidae* activity that might be used as scaffolds for the development of new and safe treatments for leishmaniasis and trypanosomiases.

## 2. Research Methods

In this study, a comprehensive literature review on inhibitors of farnesyl diphosphate synthase and squalene synthase as potential anti-*Trypanosomatidae* hit compounds is presented and discussed.

### 2.1. Literature Search

Relevant information was obtained from published and unpublished materials, with an emphasis on natural products that are potentially active against leishmaniasis and American and African trypanosomiases. Databases, such as SciFinder, Science Direct, Web of Science, PubMed (National Library of Medicine), Wiley, Scopus, the American Chemical Society (ACS), Springer, and Google Scholar, as well as theses, dissertations, and textbooks, were searched from their respective inception until April 2023 to obtain the relevant data. The search terms included "Farnesyl diphosphate synthase"; "Squalene synthase"; "Farnesyl diphosphate synthase" AND "*Trypanosoma* spp." OR "Squalene synthase" AND "*Trypanosoma* spp." OR "Farnesyl diphosphate synthase" AND "*Leishmania* spp." OR "Squalene synthase" AND "*Leishmania* spp.". Moreover, books, reports theses, and dissertations from classic literature; unpublished materials, and articles published in peer-reviewed journals related to inhibitors of squalene synthase and farnesyl diphosphate synthase, as well as leishmaniasis and American and African trypanosomiases, were also examined and searched. Furthermore, the list of references obtained from literature explorations was also searched to generate other pertinent data.

### 2.2. Data Extraction and Synthesis

Notwithstanding the database used, the full texts of potentially adequate papers were evaluated. Based on the inclusion and exclusion criteria, study selection and extraction of relevant information were unanimously carried out by both authors. Tables were used to summarize the extracted data, whereas a brief narrative description was employed to present the summing-up of the results. Graphical expression was used to present the chemical structures of active potential inhibitors of farnesyl diphosphate synthase and squalene synthase.

### 2.3. Results of the Literature Search

From the database searches, 1225 ("Squalene synthase and *Trypanosoma*": 342; "Farnesyl diphosphate synthase and *Trypanosoma*": 247; "Squalene synthase and *Trypanosoma*": 342; "Squalene synthase and *Leishmania*": 323) potentially relevant records were identified, from which 1190 were excluded after screening the titles or abstracts. Thirty-five full-length research articles were exploited to gather relevant information. In addition, data from dis-

sertations, theses, books, reports, and unpublished materials were also included. Eventually, among the inhibitors of squalene synthase and farnesyl diphosphate synthase embodied in selected documents, only those having at least micromolar inhibition, preferably in the nanomolar range, were considered in this study.

## 3. Human African Trypanosomiasis

Depending on the subspecies of the parasite involved, human African trypanosomiasis takes two forms, including *Trypanosoma brucei* rhodesiense and *Trypanosoma brucei* gambiense. This ailment is transmitted by the tsetse fly (Glossina species), which is found only in sub-Saharan Africa [15,16]. The subspecies *T.b.* gambiense is essentially responsible for chronic anthroponotic infections in Central and West Africa, accounting for approximately 95% of all HAT cases [16]. Fever, extreme fatigue, severe headaches, a skin rash, swollen lymph nodes, irritability, and aching muscles and joints are common symptoms of human African trypanosomiasis. Personality changes, progressive confusion, and other neurologic problems occur when the infection invades the central nervous system [17]. Within the circulatory system of mammalian hosts, the parasite proliferates in structurally slim and very active forms. By contrast, a nonproliferative developmental form (stumpy form) is also differentiated with the increase in parasites' numbers in the bloodstream. The latter forms are the division-interrupted forms required for tsetse flies' transmission. The procyclic forms, which are generated during the blood meal of the tsetse flies, proliferate in their midgut and further migrate to their salivary gland, whereby they differentiate into proliferative epimastigotes and attach through their flagellum. In the end, nonproliferative metacyclic forms, which are prepared for parasite transmission to the next mammalian host, are produced by the epimastigote forms [18]. For treatment, suramin and pentamidine have historically been used as the drugs of choice to treat blood-stage rhodesiense and gambiense forms, respectively. For the treatment of second-stage infections, drugs such as melarsoprol (an effective medicine used for both forms of HAT in the past) that cross the blood-brain barrier are needed. Combination therapy with eflornithine and nifurtimox has long been used as the treatment of choice to cure second-stage infections [16]. Preventive measures include wearing protective clothing, including long-sleeved shirts and pants, using mosquito repellents, and avoiding tsetse flies' bites [5].

## 4. Human American Trypanosomiasis

*Trypanosoma cruzi* is the pathogen responsible for human American trypanosomiasis (also referred to as Chagas disease). This protozoan parasite has a complex life cycle that implicates both the vectors (triatomine bugs or kissing bugs) and reservoirs (animal and human hosts) [4]. Transmission to humans occurs when humans come into contact with the urine and/or feces of infected blood-sucking triatomine bugs. In their ecological biotope, triatomine bugs become active at night, defecating close to the skin area that they have bitten. Next, the bug's waste can be inadvertently smeared into the bite or another skin break, including the mouth and eyes. Food contaminated with waste from infected triatomine bugs can typically infect groups of people with higher frequency and severity. After being bitten by a triatomine bug, a skin lesion or a purplish swelling of the lids of one eye are among the characteristic first visible signs. In addition, they can present with headache, fever, muscle pain, difficulty breathing, enlarged lymph glands, abdominal or chest pain, pallor, and swelling. During the chronic phase, the parasites are hidden mainly in the digestive muscle and heart, which might lead to destruction of the heart muscle and nervous system, resulting in cardiac arrhythmias, progressive heart failure, and sudden death. There are estimates pointing out 6–7 million people that are infected with *T. cruzi* worldwide annually [4]. This disease manifests itself in two phases, including the acute phase and the chronic phase. During the acute phase, symptoms are mild or absent and nonspecific, although a high number of parasites are in circulation in the blood [19]. The diagnosis and treatment of Chagas disease remain challenging [20]. There is a lack of effective vaccines against *T. cruzi* [21]. Early treatment is critical to manage new cases

(28,000 cases) of AT estimated to occur every year [4]. In the acute stage of the disease, nifurtimox and benznidazole (nitroimidazole compounds) are the most effective drugs used; however, in the chronic stage, their effectiveness is limited, and their use is still a matter of debate [22]. In any case, the disease should be treated in the acute stage to avoid complications that may arise in the chronic phase. To prevent this disease, it is necessary to avoid receiving blood transfusions and organ transplants in areas where Chagas disease is found, sleep under a bed net treated with insecticide, and practice safe food and water precautions [23].

## 5. Leishmaniases

Diseases that are caused by protozoan parasites from the genus *Leishmania* (including more than 20 *Leishmania* species) are called leishmaniases. These parasites are transmitted to humans through bites by infected female phlebotomine sandflies, tiny (2–3 mm) long insect vectors [6]. Mucocutaneous (MCL), cutaneous (CL), and visceral leishmaniasis (VL) are the three main forms of this disease. As most infected people do not develop disease symptoms, leishmaniasis should refer to the term of becoming sick owing to *Leishmania* infection without being infected with the parasite [6]. Today, more than 1 billion people living in areas endemic for leishmaniasis are at risk of infection. There are estimates pointing out 30,000 new cases of VL and more than 1 million new cases of CL annually [6].

In cutaneous leishmaniasis, exposed parts of the body, such as the arms, legs, and face, are affected by ulcers that produce many lesions that cause serious disability. Mainly seen in the Eastern hemisphere, Old World cutaneous leishmaniasis is caused by *L. infantum*, *L. donovani*, *L. aethiopica*, *L. tropica*, and *L. major*, whereas New World cutaneous leishmaniasis (prevalent in Central and South America) is predominantly caused by *L. amazonensis*, *L. braziliensis*, *L. guyanensis*, *L. panamensis*, *L. Mexicana*, and *L. peruviana* [24]. In the case of mucocutaneous leishmaniasis, the mucous membranes of the mouth, nose, and throat cavities are affected. In visceral leishmaniasis, symptoms such as anemia, fever, weight loss, and swelling of the spleen and liver are observed [6]. Currently used antileishmanial treatments include pentavalent antimonials [25], amphotericin B [26], miltefosine [27], paromomycin [28], and topical and systemic azoles [29]. It is difficult to deliver a complete cure for all forms of leishmaniasis, including CL, ML, and VL. Consequently, there is a tchance of relapse as the parasite remains in the body of the host. To complete their life cycle, the *Leishmania* parasites encompass vertebrate (animal with spinal cords) and invertebrate (spineless animals) host species, as well as two developmental forms (promastigotes and amastigotes sited in the sand flies and mammalian hosts, respectively) [30]. Early diagnosis and effective and prompt treatment reduce the prevalence of the disease and prevent disabilities and death. Vector control helps to interrupt or reduce the transmission of disease by reducing the number of sandflies. Control measures include the use of insecticide-treated nets, insecticide spray, environmental management, and personal protection [6].

## 6. Squalene Synthase and Farnesyl Diphosphate Synthase: Targets for Anti-*Trypanosomatidae* Drug Discovery

Squalene synthase is an enzyme that catalyzes the first reaction of the branch of the isoprenoid metabolic pathway committed specifically to sterol biosynthesis [31]. In *Leishmania* parasites, the dimerization of two farnesyl pyrophosphate molecules to produce squalene is catalyzed by squalene synthase. Next, squalene undergoes a number of enzymatic reaction steps to yield ergosterol, in contrast to cholesterol in humans [32]. The basic structure of sterol molecules (such as cholesterol, ergosterol, etc.) is derived from acetyl-CoA through the mevalonate pathway (Figure 2) [33]. In the mevalonate pathway, there is condensation of two acetyl-CoA units to afford acetoacetyl-CoA in the presence of acetoacetyl-CoA thiolase (a cytosolic enzyme). This reaction is followed by the addition of a third acetyl-CoA unit to form 3-hydroxy-3-methylglutaryl-CoA (HMG-CoA), which is then reduced by NADPH to give mevalonic acid by using two mitochondrial enzymes in

trypanosomatids, i.e., HMG-CoA synthase and HMG-CoA reductase, respectively [34]. In humans, the enzymes involved in the mevalonate pathways have been reported to occur in the cytosol and endoplasmic reticulum, especially in peroxisomes and glycosomes [33].

**Figure 2.** The mevalonate pathway to obtain mevalonic acid from acetyl-SCoA.

The isoprenoid pathway is initiated with the conversion of mevalonate into IPP (isopentenyl diphosphate) by two phosphorylation reactions followed by one decarboxylation. Afterward, dimethylallyl diphosphate (DMAPP) is obtained from the isomerization of IPP by isopentenyl diphosphate isomerase. Furthermore, condensation of IPP with DMAPP and geranyl diphosphate (GPP) generates longer chains of isoprenoids (15 carbons), namely, farnesyl diphosphate (FPP), in two successive reactions catalyzed by farnesyl diphosphate synthase (FPPS). Farnesyl diphosphate is the direct precursor for the synthesis of sterols, dichols, prenylated proteins, etc. It is noteworthy that the enzymes required for the isoprenoid pathway are located in the cytosol (*L. major*), peroxisome (animals), and mitochondria, among others [33,35]. After the isoprenoid pathway, the next two reactions comprise the first committed step in sterol biosynthesis. Next to the isoprenoid pathway, there is a condensation of two molecules of farnesyl diphosphate that generate squalene in the presence of the enzyme squalene synthase. The plausible localization of squalene synthase in *T. cruzi* and *Leishmania* spp. is either in mitochondria, microsomes, or glycosomes [33]. Afterward, squalene epoxidase catalyzes the conversion of squalene to squalene epoxide, and this reaction mostly occurs in microsomes. Next, a 2,2-oxidosqualene cyclase cyclizes the 2,3-oxidosqualene to afford lanosterol, the initial precursor of all steroid structures formed

by trypanosomatids, mammals, etc. This reaction is followed by a series of transformations to form cholesterol in mammals and ergosterol in trypanosomatids (Figure 3) [33]. The addition of a methyl group at the C24 position in the sterol side chain is one of the crucial stages of ergosterol biosynthesis that does not exist in the synthesis of cholesterol.

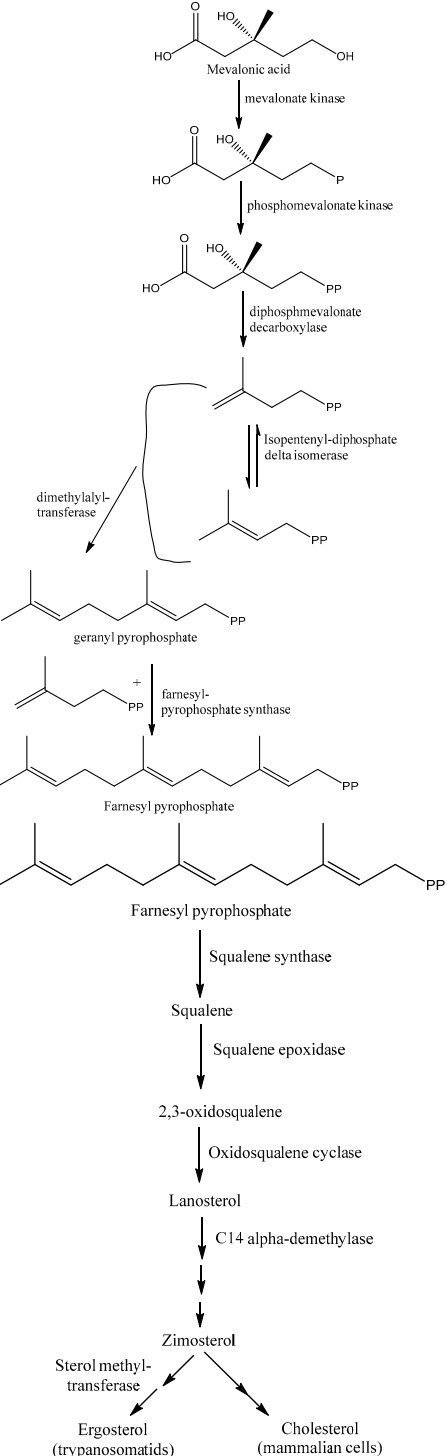

**Figure 3.** Biosynthesis of cholesterol/ergosterol from mevalonic acid (isoprenoid and squalene pathways).

## 7. Potential Inhibitors of Squalene Synthase

Squalene synthase is an attractive target mainly because it catalyzes the conversion of farnesyl diphosphate to squalene in the first committed step of cholesterol biosynthesis [36]. Inhibitors of squalene synthase include alkoxy-aminobenzhydrol

derivatives and fungus-derived zaragozic acids, but due to unfavorable toxicity profiles, none are in clinical use [37,38]. Thus, there is still a need to search for new and safe inhibitors of squalene synthase.

In the last three decades, numerous studies have been reported on the identification of squalene synthase inhibitors, and a total of forty hit compounds (Figures 4–7) are summarized in this subsection. An example is the study by Baxter et al. [39], which described the potential of squalestatin-1 (**1**) as a potent and selective inhibitor of squalene synthase in vitro with an $IC_{50}$ value of 12 nM [39,40]. Four years later, Ward et al. [41] demonstrated that a compound [3-(biphenyl-4-yl)-quinuclidine (**2**)] from the quinuclidine group exhibited in vitro inhibition of squalene synthase from human and marmoset liver microsomes with Ki values of 5 nM and 1300 nM, respectively. Moreover, McTaggart et al. [42] confirmed the inhibitory potential of 3-(biphenyl-4-yl) quinuclidine (BPQ) (**2**) and 3-(biphenyl-4-yl)-3-hydroxyquinuclidine (BPQ-OH) (**3**) by reporting their apparent inhibition constants ($K'_i$) as 12 nM and 15 nM, respectively [42]. Likewise, zaragozic acid A (**4**) has also been reported as a potent inhibitor (a competitive inhibitor with a Ki value of <1.6 nM) of squalene synthase [43–45]. 3-Hydroxy-3-[4-(quinolin-6-yl)phenyl]-1-azabicyclo [2-2-2]octane dihydrochloride (**5**) is a potent inhibitor of rat liver microsomal squalene synthase, with an $IC_{50}$ value less than 0.9 nM. Under in vivo conditions, oral administration of this compound (3-hydroxy-3-[4-(quinolin-6-yl)phenyl]-1-azabicyclo [2-2-2]octane dihydrochloride) (**5**) to rats inhibited de novo [14C]cholesterol biosynthesis from [14C]mevalonate in the liver with an $ED_{50}$ value of 5 mg/kg [46]. A quinuclidin derivative ((E)-2-[2-uoro-2-(quinuclidin-3-ylidene) ethoxy]-9H-carbazole monohydrochloride) (**6**) was reported to inhibit squalene synthase from hepatic microsomes of several species, including rats, hamsters, guinea pigs, rhesus monkeys, and HepG2 cells ($IC_{50}$: 90, 170, 46, 45, and 79 nM, respectively). Under in vivo conditions, this compound equally inhibited squalene synthase activities in hepatic microsomes and suppressed cholesterol biosynthesis in rats ($ED_{50}$: 32 mg/kg) [47].

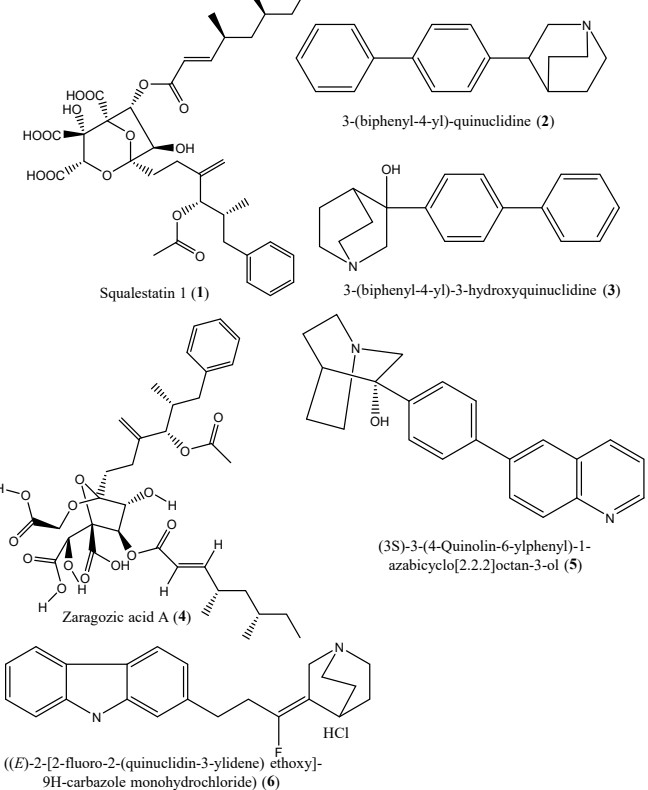

**Figure 4.** Chemical structures of the squalene synthase inhibiors (**1–6**).

5-{N-[2-Butenyl-3-(2-methoxyphenyl)]-N-methylamino}-1,1-penthylidenebis(phosphonic acid) trisodium salt (**7**)

(**8**): Tripivaloyloxymethyl derivative of compound **7**

Compound (**9**)

|      |                            |
|------|----------------------------|
| (**10**) | R=Ac                   |
| (**11**) | R = COCH$_2$CH$_3$     |
| (**12**) | R = CO(CH$_2$)$_2$CH$_3$ |
| (**13**) | R = COCH$_2$(CH$_3$)$_2$ |
| (**14**) | R = CH$_2$OCOC(CH$_3$)$_3$ |

1-[[(3R,5S)-7-Chloro-5-(2,3-dimethoxyphenyl)-1-(3-hydroxy-2,2-dimethylpropyl)-2-oxo-1,2,3,5-tetrahydro-4,1-benzoxazepin-3-yl]acetyl]piperidine-4-acetic acid (**9**) and [1-[[(3R,5S)-1-(3-Acetoxy-2,2-dimethylpropyl)-7-chloro5-(2,3-dimethoxyphenyl)-2-oxo-1,2,3,5-tetrahydro-4,1-benzoxazepin-3-yl]acetyl]piperidine-4-acetic acid (**10**); 1-[[(3R,5S)-7-chloro-5-(2,3-dimethoxyphenyl)-1-[2,2-dimethyl-3-(propionyloxy)propyl]-2-oxo-1,2,3,5-tetrahydro-4,1-benzoxazepin-3-yl]acetyl]piperidine-4-acetic acid (**11**); 1-[[(3R,5S)-1-[3-(Butyryloxy)-2,2-dimethylpropyl]-7-chloro-5-(2,3-dimethoxyphenyl)-2-oxo-1,2,3,5-tetrahydro4,1-benzoxazepin-3-yl]acetyl]piperidine-4-acetic acid (**12**); 1-[[(3R,5S)-7-chloro-5-(2,3-dimethoxyphenyl)-1-[2,2-dimethyl-3-(isobutyryloxy)propyl]-2-oxo-1,2,3,5-tetrahydro-4,1-benzoxazepin-3-yl]acetyl]piperidine-4-acetic acid (**13**); 1-[[(3R,5S)-7-chloro-5-(2,3-dimethoxyphenyl)-1-[2,2-dimethyl-3-(pivaloyloxymethyloxy)propyl]-2-oxo-1,2,3,5-tetrahydro-4,1-benzoxazepin-3-yl]acetyl]piperidine-4-acetic acid (**14**)]

**Figure 5.** Chemical structures of compounds **7**–**14** as potential squalene synthase inhibitors.

Zaragozic acids and derivatives of quinuclidine are well-known inhibitors of squalene synthase; however, due to their unfavorable toxicity profiles, these promising compounds have failed to enter the clinical trial phase [38]. Furthermore, computer simulations indicated strong interactions between squalene synthase and zaragozic acid with a high docking score, confirming the in vitro enzymatic results [38].

Several nitrogen- and nonnitrogen-containing bisphosphonate derivatives (BPs) were reported as inhibitors of squalene synthase. One such example includes 5-(N-[2-butenyl-3-(2-methoxyphenyl)]-N-methylamino)-1,1-penthylidenebis(phosphonic acid) trisodium salt (**7**) (IC$_{50}$: 3.6 nM), which selectively inhibited the activity of squalene synthase. In addition, a tripivaloyloxymethyl derivative (**8**) of compound **7** demonstrated inhibitory potential vis-à-vis squalene synthase with an IC$_{50}$ value of 39 μM [48]. However, these nitrogen-containing BPs are not metabolized and are excreted unaffected via the kidney. Notably, BPs are typically administered under fasting conditions, as food reduces their bioavailability [49].

[(Z)-3-[2-(9H-Fluoren-2-yloxy)ethylidene]quinuclidine hydrochloride (**15**)

(E)-3-[2-(9H-fluoren-2-yloxy)ethylidene]quinuclidine hydrochloride (**16**)

3-[2-(9H-fluoren-2-yloxy)ethyl]quinuclidine hydrochloride (**17**)

(Z)-3-[2-(9H-fluoren-2-ylamino)ethylidene]
quinuclidine hydrochloride (**18**)

(Z)-3-[2-[N-(9H-fluoren-2-yl)-N-methylamino]ethylidene]-
quinuclidine fumarate (**19**)

(Z)-2-[2-Quinuclidin-3-ylidene)ethoxy]-9H-carbazole
hydrochloride (**20**)

(E)-2-[2-Fluoro-2-(quinuclidin-3-ylidene)ethoxy]-9H-carbazole hydrochloride (**21**)

{(3R)-3-[[2-benzyl-6-[3R,4S]-3-hydroxy-4-methoxypyrrolidin-1-
yl]pyridin-3-yl]ethynyl]quinuclidin-3-ol monohydrate} (**22**)

3-(Biphenyl-4-yl)-3-hydroxy
quinuclidine (**24**)

3-(Biphenyl-4-yl)-2,3-dehydroquinuclidine
(**25**)

{(3R)-3-[[2-benzyl-6-(3-methoxypropyloxy)-pyridin-3-
yl]ethynyl]quinuclidin-3-ol} (**23**)

3-(Biphenyl-4-yl-40-hydroxy)-2,3-
dehydroquinuclidine (**26**)

**Figure 6.** Chemical structures of the potential squalene inhibitors **15–26**.

2-(4-Biphenyl)-4-methyl-octahydro-1,4-
benzoxazin-2-ol, hydrobromide (**27**)

2-(4-biphenyl)-2-(3-nitrooxypropoxy)-4-
methylmorpholine, hydrobromide (**28**)

29,  R= n-butyl
30,  R= n-pentyl
31,  R= n-hexyl
32,  R= n-heptyl
33,  R= 3-methyl-n-butyl
34,  R= cyclohexyl

(*R*)-5-(5-(4-methoxy-3-methylphenyl)
furan-2-yl)-2,2-dimethyl-2,3,5,6-
tetrahydrobenzo[*a*]phenanthridin-4(1*H*)-one (**35**)

(*R*)-*N*-(2-(2-((3-methoxybenzyl)oxy)phenyl)-1,4
-dihydroquinazolin-3(2*H*)-
yl)benzo[*d*][1,3]dioxole-5-carboxamide (**36**)

(2*S*,3*R*)-8-((*R*)-1-(3,5-dihydroxyphenyl)ethyl)-3-hydroxy-
2-methyl-2-(naphthalen-2-ylmethyl)-3,4,7,8-tetrahydro-
2*H*,6*H*-pyrano[3,2-*g*]chromen-6-one (**37**)

2-(2-(7-chloro-5-(naphthalen-1-yl)-1-neopentyl-2-oxo-
1,2,3,5-tetrahydrobenzo[*e*][1,4]oxazepin-3-yl)acetamido)
butanoic acid (**38**)

Carnosol (**39**)

3'-O-Methyldiplacol (**40**)

**Figure 7.** Chemical structures of potential squalene synthase inhibitors **27–40**.

In another experiment, a series of 4,1-benzoxazepine derivatives [1-[[(3R,5S)-7-chloro-5-(2,3-dimethoxyphenyl)-1-(3-hydroxy-2,2-dimethylpropyl)-2-oxo-1,2,3,5-tetrahydro-4,1-benzoxazepin-3-yl]acetyl]piperidine-4-acetic acid (**9**); 1-[[(3R,5S)-1-(3-acetoxy-2,2-dimethylpropyl)-7-chloro5-(2,3-dimethoxyphenyl)-2-oxo-1,2,3,5-tetrahydro-4,1-benzoxazepin-3-yl]acetyl]piperidine-4-acetic acid (**10**); 1-[[(3R,5S)-7-chloro-5-(2,3-dimethoxyphenyl)-1-[2,2-dimethyl-3-(propionyloxy)propyl]-2-oxo-1,2,3,5-tetrahydro-4,1-benzoxazepin-3-yl]acetyl]piperidine-4-acetic acid (**11**); 1-[[(3R,5S)-1-[3-(butyryloxy)-2,2-dimethylpropyl]-7-chloro-5-(2,3-dimethoxyphenyl)-2-oxo-1,2,3,5-tetrahydro4,1-benzoxazepin-3-yl]acetyl]piperidine-4-acetic acid (**12**); 1-[[(3R,5S)-7-chloro-5-(2,3-dimethoxyphenyl)-1-[2,2-dimethyl-3-(isobutyryloxy)propyl]-2-oxo-1,2,3,5-tetrahydro-4,1-benzoxazepin-3-yl]acetyl]piperidine-4-acetic acid (**13**); 1-[[(3R,5S)-7-chloro-5-(2,3-dimethoxyphenyl)-1-[2,2-dimethyl-3-(pivaloyloxymethyloxy)propyl]-2-oxo-1,2,3,5-tetrahydro-4,1-benzoxazepin-3-yl]acetyl]piperidine-4-acetic acid (**14**)] were synthesized and evaluated for inhibition of squalene synthase and sterol synthesis in rat liver by Miki et al. [50]. Under in vitro conditions, compounds **9–14** inhibited squalene synthase with $IC_{50}$ values of 45, 76, 87, 93, 89 and 471 nM, respectively. Under in vivo conditions, the single oral administration of 10 mg/kg compounds **9–14** to [14C]acetate (50 $\mu$Ci/kg)-induced Wistar rats inhibited sterol synthesis by 64, 81, 40, 30, 34, and 43%, respectively [50].

Although the exact mechanism of action of benzoxazepin derivatives has not yet been fully elucidated, they are believed to interact with multiple receptors, including serotonin, dopamine, and norepinephrine receptors [51], modulate the activity of certain enzymes, such as monoamine oxidase [52], or interact with certain ion channels, such as calcium and potassium channels [51,53]. Furthermore, Urbina et al. [10] verified and confirmed the *T. cruzi* epimastigote squalene synthase inhibitory (noncompetitive inhibitor) potential of 3-(biphenyl-4-yl)-3-hydroxyquinuclidine (3-biphenyl4-yl-1-aza-bicyclo [2.2.2]-octan-3-ol; BPQ-OH) (**3**) with a Ki value of 48 nM. Furthermore, in vitro anti-*T. cruzi* (amostigote forms) ($IC_{50}$: 3 $\mu$M; SI: 33.33, Vero cells) and anti-*L. mexicana* (promastigotes) ($IC_{50}$: 12 $\mu$M) experiments revealed low $IC_{50}$ values. Although appropriate controls were not used for the assessment, the results of this study clearly indicated the selective antiparasitic activity of the quinuclidine derivative BPQ-OH. A year later, Ishihara et al. [54] reported a series of five 3-ethylidenequinuclidine derivatives (compounds **15–19**) [(Z)-3-[2-(9H-Fluoren-2-yloxy)ethylidene]quinuclidine hydrochloride (**15**); (E)-3-[2-(9H-fluoren-2-yloxy)ethylidene]quinuclidine hydrochloride (**16**); 3-[2-(9H-fluoren-2-yloxy)ethyl]quinuclidine hydrochloride (**17**); (Z)-3-[2-(9H-fluoren-2-ylamino)ethylidene]quinuclidine hydrochloride (**18**); and (Z)-3-[2-[N-(9H-fluoren-2-yl)-N-methylamino]ethylidene]quinuclidine fumarate (**19**)] that inhibits squalene synthase from the hamster liver with $IC_{50}$ values of 0.076, 0.15, 0.25, 0.27 and 0.56 $\mu$M, respectively. Compound (**15**), which was the most potent inhibitor, also demonstrated effective inhibition against human hepatoma cells ($IC_{50}$: 48 nM). This research group also found that (Z)-2-[2-(quinuclidin-3-ylidene)ethoxy]-9H-carbazole hydrochloride (**20**) and (E)-2-[2-fluoro-2-(quinuclidin-3-ylidene)ethoxy]-9H-carbazole hydrochloride (**21**) inhibit squalene synthase derived from human hepatoma cells, with $IC_{50}$ values of 160 and 79 nM, respectively [55]. In 2004, Urbina et al. [11] prepared two other quinuclidine-based derivatives [{(3R)-3-[[2-benzyl-6-[(3R,4S)-3-hydroxy-4-methoxypyrrolidin-1-yl]pyridin-3-yl]ethynyl]quinuclidin-3-ol monohydrate} (**22**) and {(3R)-3-[[2-benzyl-6-(3-methoxypropyloxy)pyridin-3-yl]ethynyl]quinuclidin-3-ol} (**23**)], which exhibited noncompetitive inhibition toward *T. cruzi* proteins with $IC_{50}$ values of 5.4 and 7.2 nM, respectively, for glycosomal squalene synthase and 15 and 5.5 nM, respectively, for microsomal squalene synthase of *T. cruzi* epimastigotes. To verify the direct effect on the parasite in culture, further screening of compounds (**22**) and (**23**) afforded $IC_{50}$ values of 10 nM and 0.4 to 1.6 nM against extracellular epimastigotes and intracellular amastigotes, respectively. Under in vivo conditions, **22** was able to provide full protection against death and completely arrest the development of parasitemia when given at a dose of 50 mg/kg of body weight for 30 days [11,56]. Other biphenyl quinuclidine derivatives (**24–26**) were also evaluated for their inhibitory activity toward *L. major* squalene synthase. Compounds (3-(biphenyl-4-yl)-3-hydroxyquinuclidine) (**24**), (3-(biphenyl-4-yl)-2,3-dehydro-quinuclidine)

(**25**), and (3-(biphenyl-4-yl-40-hydroxy)-2,3-dehydro-quinuclidine) (**26**) afforded IC$_{50}$ values of 0.013, 0.243, and 0.096 µM, respectively. Against *L. donovani* intracellular amastigotes, compounds (**24–26**) afforded ED$_{50}$ values of 29.0, 74.3 and >108 µM, respectively, attesting to the enzymatic effectiveness of these compounds against *L. major* squalene synthase [57].

In 2006, Tavridou et al. [58] reported the inhibitory potential of two biphenyl derivatives, i.e., 2-(4-Biphenyl)-4-methyl-octahydro-1,4-benzoxazin-2-ol, hydrobromide (**27**) and 2-(4-biphenyl)-2-(3-nitrooxy propoxy)-4-methylmorpholine, hydrobromide (**28**) on squalene synthase in rabbit liver microsomes with IC$_{50}$ values of 33 and 0.6 µM, respectively. In human liver microsomes, compounds (**27** and **28**) afforded IC$_{50}$ values of 63 and 1 µM, respectively. Simvastatin (a standard squalene synthase inhibitor; IC$_{50}$: 30 µM) inhibits the activity of the rabbit enzyme by 23% but had poor or no effect on the activity of the human enzyme (percent inhibition: 19% at 100 µM) [58]. Developed as cholesterol-lowering agents, two arylquinuclidine derivatives {(3R)-3-[[2-benzyl-6-(3-methoxypropyloxy)-pyridin-3-yl]ethynyl]quinuclidin-3-ol} (ER-119884) (**23**) and {(3R)-3-[[2-benzyl-6-[3R,4S)-3-hydroxy-4-methoxypyrrolidin-1-yl]pyridin-3-yl]ethynyl]quinuclidin-3-ol monohydrate} (**22**) were found to be potent noncompetitive inhibitors of native *L. amazonensis* squalene synthase (glycosomal SQS: Ki = 6.4 and 6.9 nM for compounds (**23** and **22**), respectively; microsomal or mitochondrial SQS: Ki = 5.5 and 14.8 nM for compounds (**23** and **22**), respectively). Furthermore, the antileishmanial activity of these compounds revealed IC$_{50}$ values in the nanomolar range when tested against promastigotes (IC$_{50}$ value: 14.7 and 1.7 nM for compounds (**23** and **22**), respectively) and intracellular amastigotes (IC$_{50}$ value: 4.0 and 0.9 nM for compounds (**23** and **22**), respectively) (after 72 h of incubation). Growth inhibition was definitely associated with the depletion of the parasite's endogenous sterols and the concomitant accumulation of exogenous cholesterol [59].

In addition, Rodrígues-Poveda et al. [9] synthesized a series of 2-alkyl aminoethyl-1,1-bisphosphonic acids (compounds **9–34**) as potent inhibitors of *T. cruzi* squalene synthase with IC$_{50}$ values of 39.0, 5.0, 21.4, 11.9, 22.0, and 30.0 nM for compounds (**29–34**), respectively. Further in vitro studies demonstrated that compounds **29–33** inhibit the growth of *T. cruzi* amastigotes with IC$_{50}$ values of 4.8, 0.54, 0.84, 10.0, and 0.94 nM, respectively [9]. Huang et al. [60] described the in vitro squalene synthase inhibitory activity of four compounds, including compounds (**35–38**), with IC$_{50}$ values of 1.7, 0.14, 191, and 63 nM, respectively. More recently, Macías-Alonso et al. [61] reported the inhibitory effect of an abietane-type diterpenoid (**39**) (carnosol; IC$_{50}$: 17.6 µM) isolated from cultivated *Salvia canariensis*. In a computational study by Wadanambi and Mannapperuma [32], 3-O-methyl diplacol (**40**) strongly bound to *Leishmania donovani* squalene synthase with a binding energy of −9.00 kcal/mol (vs. ancistrotanzanine B: −9.83 kcal/mol), inferring that this compound possesses squalene synthase inhibitory action [32].

From 1992 to 2021, a total of 40 compounds (Table 1, Figures 4–7) were reported to inhibit the enzyme squalene synthase in in vitro and in vivo studies. Although diseases caused by *Trypanosomatidae* have several treatment options, the therapy displays many problems, such as extensive toxicity, a parenteral route of administration, and a lack of efficacy, affecting compliance, high costs, and emerging drug resistance. In the last few decades, success in anti-*Trypanosomatidae* drug discovery has been acceptable; however, almost no further research beyond the academic results has been achieved.

**Table 1.** Prospective inhibitors of squalene synthase.

| Serial Number | Compound Name | Model | Significant Results | References |
|---|---|---|---|---|
| 1 | Squalestatin-1 (**1**) | SQS in rat liver microsomes | $IC_{50}$: 12 nM | [39] |
| 2 | 3-(biphenyl-4-yl)-quinuclidine (**2**) | Microsomal SQS from human or marmoset liver | Ki: 5 nM and 1300 nM in microsomes prepared from human and marmoset liver, respectively | [40] |
| 3 & 4 | 3-(biphenyl-4-yl) quinuclidine (BPQ) (**2**) and 3-(biphenyl-4-yl)-3-hydroxyquinuclidine (BPQ-OH) (**3**) | *T. cruzi* epimastigote's squalene synthase; NS | Ki: 12 nM and 15 nM; Ki value: 48 nM | [10,42] |
| 5 | 3-hydroxy-3-[4-(quinolin-6-yl)phenyl]-1-azabicyclo[2-2-2]octane dihydrochloride (**5**) | Rat liver microsomal squalene synthase | - In vitro studies: $IC_{50} < 0.9$ nM; <br> - In vivo studies: <br><br> Reduction of cholesterol synthesis with $ED_{50}$ value of 5 mg/kg | [46] |
| 6 | ((E)-2-[2-uoro-2-(quinuclidin-3-ylidene) ethoxy]-9H-carbazole monohydrochloride) (**6**) | Hepatic microsomes of rats, Hamster, guinea pig, Rhesus monkey and HepG2 cells | In vitro studies: $IC_{50}$: 90, 170, 46, 45, and 79 nM, respectively; In vivo studies: Reduction of cholesterol synthesis with $ED_{50}$: 32 mg/kg | [47] |
| 7 & 8 | 5-(N-[2-butenyl-3-(2-methoxyphenyl)]-N-methylamino)-1,1-penthylidenebis(phosphonic acid) trisodium salt (**7**) and tripivaloyloxymethyl derivative (**8**) | SQS in SD (Sprague Dawley) rat liver microsomes | $IC_{50}$: 3.6 nM and 39 µM, respectively | [48] |
| 9–14 | 1-[[(3R,5S)-7-Chloro-5-(2,3-dimethoxyphenyl)-1-(3-hydroxy-2,2-dimethylpropyl)-2-oxo-1,2,3,5-tetrahydro-4,1-benzoxazepin-3-yl]acetyl]piperidine-4-acetic acid (**9**); 1-[[(3R,5S)-1-(3-Acetoxy-2,2-dimethylpropyl)-7-chloro5-(2,3-dimethoxyphenyl)-2-oxo-1,2,3,5-tetrahydro-4,1-benzoxazepin-3-yl]acetyl]piperidine-4-acetic acid (**10**); 1-[[(3R,5S)-7-chloro-5-(2,3-dimethoxyphenyl)-1-[2,2-dimethyl-3-(propionyloxy)propyl]-2-oxo-1,2,3,5-tetrahydro-4,1-benzoxazepin-3-yl]acetyl]piperidine-4-acetic acid (**11**); 1-[[(3R,5S)-1-[3-(Butyryloxy)-2,2-dimethylpropyl]-7-chloro-5-(2,3-dimethoxyphenyl)-2-oxo-1,2,3,5-tetrahydro4,1-benzoxazepin-3-yl]acetyl]piperidine-4-acetic acid (**12**); 1-[[(3R,5S)-7-chloro-5-(2,3-dimethoxyphenyl)-1-[2,2-dimethyl-3-(isobutyryloxy)propyl]-2-oxo-1,2,3,5-tetrahydro-4,1-benzoxazepin-3-yl]acetyl]piperidine-4-acetic acid (**13**); 1-[[(3R,5S)-7-chloro-5-(2,3-dimethoxyphenyl)-1-[2,2-dimethyl-3-(pivaloyloxymethyloxy)propyl]-2-oxo-1,2,3,5-tetrahydro-4,1-benzoxazepin-3-yl]acetyl]piperidine-4-acetic acid (**14**)] | Squalene synthase of rat liver | $IC_{50}$ values: 45, 76, 87, 93, 89 and 471 nM, respectively. | [50] |

**Table 1.** *Cont.*

| Serial Number | Compound Name | Model | Significant Results | References |
|---|---|---|---|---|
| 15–19 | [(Z)-3-[2-(9H-Fluoren-2-yloxy)ethylidene]quinuclidine hydrochloride (**15**); (E)-3-[2-(9H-fluoren-2-yloxy)ethylidene]quinuclidine hydrochloride (**16**); 3-[2-(9H-fluoren-2-yloxy)ethyl]quinuclidine hydrochloride (**17**); (Z)-3-[2-(9H-fluoren-2-ylamino)ethylidene]quinuclidine hydrochloride (**18**); (Z)-3-[2-[N-(9H-fluoren-2-yl)-N-methylamino]ethylidene] quinuclidine fumarate (**19**) | Squalene synthase from the hamster liver | $IC_{50}$s: 0.076, 0.15, 0.25, 0.27 and 0.56 μM, for compounds **15–19**, respectively | [54] |
| 20, 21 | (Z)-2-[2-(quinuclidin-3-ylidene)ethoxy]-9H-carbazole hydrochloride (**20**) and (E)-2-[2-fluoro-2-(quinuclidin-3-ylidene)ethoxy]-9H-carbazole hydrochloride (**21**) | Squalene synthase derived from human hepatoma cells | $IC_{50}$s: 160 and 79 nM, respectively | [55] |
| 22, 23 | {(3R)-3-[[2-benzyl-6-[(3R,4S)-3-hydroxy-4-methoxypyrrolidin-1-yl]pyridin-3-yl]ethynyl]quinuclidin-3-ol monohydrate} (**22**) and {(3R)-3-[[2-benzyl-6-(3-methoxypropyloxy)pyridin-3-yl]ethynyl]quinuclidin-3-ol} (**23**) | Glycosomal and microsomal SQS of *T. cruzi* epimastigotes *L. amazonensis* squalene synthase | - $IC_{50}$ values: <br> • glycosomal SQS: 5.4 and 7.2 nM, respectively <br> • microsomal SQS: <br> $IC_{50}$ values of 15 and 5.5 nM, respectively; <br> - in vivo studies: complete arrest of the development of parasitemia when compound E5700 was given at a dose of 50 mg/kg of body weight for 30 days; <br> - $IC_{50}$s: Glycosomal SQS: 6.4 and 6.9 nM for compounds **23** and **22**, respectively; <br> - Microsomal or mitochondrial SQS: 5.5 and 14.8 nM for compounds **23** and **22**, respectively | [11,56,59] |
| 24–26 | (3-(Biphenyl-4-yl)-3-hydroxyquinuclidine) (**24**); (3-(Biphenyl-4-yl)-2,3-dehydroquinuclidine); (**25**) and 5b (3-(Biphenyl-4-yl-40-hydroxy)-2,3-dehydroquinuclidine) (**26**) | *L. major* squalene synthase | $IC_{50}$s: 0.013, 0.243 and 0.096 μM, for compounds **24–26**, respectively; In vivo studies: Against *L. donovani* intracellular amastigotes, the following $ED_{50}$ values were obtained: 29.0, 74.3 and >108 μM, respectively | [57] |

**Table 1.** *Cont.*

| Serial Number | Compound Name | Model | Significant Results | References |
|---|---|---|---|---|
| 27–28 | 2-(4-Biphenyl)-4-methyl-octahydro-1,4-benzoxazin-2-ol, hydrobromide (**27**) and 2-(4-biphenyl)-2-(3-nitrooxypropoxy)-4-methylmorpholine, hydrobromide (**28**) | Squalene synthase in rabbit liver microsomes | $IC_{50}$s: of 33 and 0.6 μM for compounds **27** and **28**, respectively | [58] |
| 29–34 | Compounds **29–33** and **34** | *T. cruzi* squalene synthase | $IC_{50}$s: 39.0, 5.0, 21.4, 11.9, 22.0 and 30.0 nM, respectively | [9] |
| 35–38 | (R)-5-(5-(4-methoxy-3-methylphenyl)furan-2-yl)-2,2-dimethyl-2,3,5,6-tetrahydrobenzo[a]phenanthridin-4(1H)-one (**35**); (R)-N-(2-(2-((3-methoxybenzyl)oxy)phenyl)-1,4-dihydroquinazolin-3(2H)-yl)benzo[d][1,3]dioxole-5-carboxamide (**36**); (2S,3R)-8-((R)-1-(3,5-dihydroxyphenyl)ethyl)-3-hydroxy-2-methyl-2-(naphthalen-2-ylmethyl)-3,4,7,8-tetrahydro-2H,6H-pyrano[3,2-g]chromen-6-one (**37**) and 2-(2-(7-chloro-5-(naphthalen-1-yl)-1-neopentyl-2-oxo-1,2,3,5-tetrahydrobenzo[e][1,4]oxazepin-3-yl)acetamido)butanoic acid (**38**) | NS | $IC_{50}$s: 1.7, 0.14, 191 and 63 nM for compounds **35–38**, respectively | [60] |
| 39 | Carnosol (**39**) | NS | $IC_{50}$: 17.6 μM | [61] |

SQS: squalene synthase.

Phenotypic drug screening and target-based drug discovery are the main approaches that have been used so far to identify pharmacologically active lead compounds against the *Leishmania* and *Trypanosoma* parasites [62]. Nevertheless, recent discoveries have demonstrated that anti-*Trypanosomatidae* drug development can be improved. It is hypothesized that the identification and validation of new disease-relevant targets might provide more insights for novel chemical scaffolds (hit discovery) or known drugs (drug repurposing). Perhaps the lead optimization might improve the activity of selected hits. Furthermore, the intensification of funding and public-private partnerships may contribute to the sustainability of anti-*Trypanosomatidae* drug discovery and beyond [63]. As both *Leishmania* and *Trypanosoma* species lack crucial enzymes that are necessary for cholesterol biosynthesis [64], while producing a particular class of non-mammalian sterols (ergosterols and 24-methyl sterols, among others), which are essential for their growth and viability, inhibiting important enzymes of the biosynthetic pathways of such compounds might afford potential targets for the discovery of new anti-*Trypanosomatidae* compounds [33]. These secondary metabolites are produced through a sequential process that includes pathways such as the isoprenoid pathway and sterol biosynthesis, which involve the key enzymes farnesyl diphosphate synthase and squalene synthase, respectively. As the production of sterols is crucial for the survival and virulence of *Trypanosomatidae*, inhibitors of squalene synthase and farnesyl diphosphate synthase can be prospected as scaffolds for the identification of anti-*Trypanosomatidae* compounds.

Among the inhibitors of squalene synthase, the in vitro activity of eight compounds was validated against *Trypanosoma cruzi* [(3-(biphenyl-4-yl)-3-hydroxyquinuclidine (3-biphenyl4-yl-1-aza-bicyclo [2.2.2]-octan-3-ol (BPQ-OH) (3); $IC_{50}$: 3 $\mu$M; SI: 33.33, Vero cells, [10]; (compounds (**29–33**)); $IC_{50}$ values: 4.8, 0.54, 0.84, 10.0, and 0.94 nM, respectively; Rodrígues-Poveda et al., 2012) [9]; (compounds **22** and **23**; $IC_{50}$ values: 10 nM and 0.4 to 1.6 nM; extracellular epimastigotes and intracellular amastigotes, respectively [11]; whereas the in vitro phenotypic screening of five compounds (**24–26**); $ED_{50}$s: 29.0, 74.3, and >108 $\mu$M, respectively; *L. donovani* intracellular amastigotes, [57]; compounds (**22 and 23**); $IC_{50}$ values: 1.7 and 14.7 nM, respectively, [59] were substantiated against *Leishmania* spp. in culture.

## 8. Potential Inhibitors of Farnesyl Diphosphate Synthase

An enzyme that belongs to the (E)-prenyltransferases, i.e., farnesyl diphosphate synthase (FPPS), catalyzes the production of farnesyl pyrophosphate (FPP; a C15 product) through the condensation of two units of isopentenyl diphosphate (IPP, C5) along with one unit of dimethylallyl diphosphate (DMAPP, C5) [65].

According to the literature, a variety of compounds (Figures 8 and 9) have been reported to inhibit the activity of farnesyl diphosphate synthase, an enzyme involved in several metabolic pathways, including sterol biosynthesis. *Trypanosomatidae* are parasite species that are responsible for leishmaniasis and trypanosomiasis in humans and strongly depend on metabolites, such as ergosterol and other proteins, for their growth and survival. It is therefore hypothesized that farnesyl diphosphate synthase is among the attractive targets for the discovery of anti-*Trypanosomatidae* hit compounds. In a previous study by Szajnman et al. [66], *Trypanosoma cruzi* farnesyl pyrophosphate synthase was used as an experimental model to assess the inhibitory potential of a series of nonnitrogen-containing bisphosphonates, including compounds (**41–50**) ($IC_{50}$: 42.83, 1.94, 2.37, 9.36, 8.45, 5.71, 5.67, 4.54, 19.73, and 4.25 $\mu$M, respectively).

(1-hydroxypentane-1,1-diyl)bis(phosphonic acid) (**41**), n=2
(1-hydroxyhexane-1,1-diyl)bis(phosphonic acid) (**42**), n=4
(1-hydroxyheptane-1,1-diyl)bis(phosphonic acid) (**43**), n=5
(1-hydroxyoctane-1,1-diyl)bis(phosphonic acid) (**44**), n=6
(1-hydroxynonane-1,1-diyl)bis(phosphonic acid) (**45**), n=7

pentane-1,1-diylbis(phosphonic acid) (**46**), n=3
hexane-1,1-diylbis(phosphonic acid) (**47**), n=4
heptane-1,1-diylbis(phosphonic acid) (**48**), n=5
octane-1,1-diylbis(phosphonic acid) (**49**), n=6
nonane-1,1-diylbis(phosphonic acid) (**50**), n=7

(1-aminopentane-1,1-diyl)bis(phosphonic acid) (**52**), n=4
(1-aminoheptane-1,1-diyl)bis(phosphonic acid) (**53**), n=5

Risedronate (**51**)

1-[(n-Oct-1-ylamino)ethyl] 1,1-bisphosphonic acid (**54**);
1-[(tert-Butylamino)ethyl] 1,1-bisphosphonic acid (**55**).

**54**, R= n-octyl
**55**, R= tert-butyl

**56**, R= n-dodecyl

1-[(n-dodec-1-ylamino)ethyl] 1,1-bisphosphonic acid (**56**)

1-[(n-isoprop-1-ylamino)ethyl] 1,1-bisphosphonic acid (**57**), R=iso-propyl
2-(n-propylamino) ethane-1,1-diyl]bisphosphonic acid (**58**), R= n-propyl

(2-(octylthio)ethane-1,1-diyl)bis(phosphonic acid) (**59**), R= n-octyl

(2-(nonylthio)ethane-1,1-diyl)bis(phosphonic acid) (**60**), R= n-nonyl

(2-(decylthio)ethane-1,1-diyl)bis(phosphonic acid) (**61**), R= n-decyl

**Figure 8.** Prospective inhibitors of farnesyl pyrophosphate synthase (compounds (**41–61**)).

$N^4,N^{4'}$-bis(3-(4,5-dihydro-1$H$-imidazol-2-yl)phenyl)-[1,1'-biphenyl]-4,4'-dicarboxamide (**62**)

$N^1,N^4$-bis(3-carbamoyl-5-(4,5-dihydro-1$H$-imidazol-2-yl)phenyl)terephthalamide (**66**)

$N^1,N^3$-bis(4-(4,5-dihydro-1$H$-imidazol-2-yl)phenyl)isophthalamide (**63**)

$N^1,N^4$-bis(3-(4,5-dihydro-1$H$-imidazol-2-yl)phenyl)terephthalamide (**67**)

2-amino-$N^1$-(4-(4-methyl-4,5-dihydro-1$\lambda^4$,3$\lambda^2$-imidazol-2-yl)phenyl)-$N^4$-(4-(4-methyl-4,5-dihydro-1$H$-imidazol-2-yl)phenyl)terephthalamide (**64**)

2-amino-$N^1,N^4$-bis(4-(4,5-dihydro-1$H$-imidazol-2-yl)phenyl)terephthalamide (**68**)

$N^1,N^4$-bis(2-(4,5-dihydro-1$H$-imidazol-2-yl)phenyl)terephthalamide (**65**)

2-amino-$N^1,N^4$-bis(3-(4,5-dihydro-1$H$-imidazol-2-yl)phenyl)terephthalamide (**69**)

2-amino-$N^1,N^3$-bis(4-(4,5-dihydro-1$H$-imidazol-2-yl)phenyl)isophthalamide (**70**)

$N^1,N^4$-bis(4-(4,5-dihydro-1$H$-imidazol-2-yl)-3-hydroxyphenyl)terephthalamide (**71**)

Quinone methide celastrol (**72**)

3-fluoro-1-(2-hydroxy2,2-diphosphono-ethyl)pyridinium (**73**)

1-(2-hydroxy-2,2-diphosphonoethyl)-3-phenylpyridinium (**74**)

**Figure 9.** Chemical structures of potent inhibitors of farnesyl diphosphate synthase (**62–74**).

Upon incubation of compounds (**41**, **43**, **45**, and **49**) with the amastigote forms of *Trypanosoma cruzi*, the inhibition was effective, yielding IC$_{50}$ values of 21.4, 18.1, 65.8, and 22.36 μM, respectively. Against *Trypanosoma brucei* farnesyl pyrophosphate synthase, compounds (**41–45**) afforded IC$_{50}$ values of 3.12, 0.66, 3.57, and 4.54 μM, respectively [66]. Nonnitrogen-containing BPs are known to induce apoptosis, as these compounds metabolize into cytotoxic, nonhydrolyzable analogs of ATP, which interfere with mitochondrial function [67–70]. Other nitrogen-containing BPs (compounds (**51–60**)) were also reported as inhibitors of farnesyl pyrophosphate synthase. Complete growth arrest of the extracellular epimastigote form of *T. cruzi* (IC$_{50}$: 150 μM) by the bisphosphonate risedronate (**51**) was attributed to the depletion of the parasite's endogenous sterols (inhibition of ergosterol synthesis with an IC$_{50}$ value of 65.4 μM) [71]. Likewise, two alkyl 1-amino-1,1-bisphosphonates (compounds (**52 and 53**)) inhibited *T. cruzi* farnesyl pyrophosphate synthase with IC$_{50}$ values of 0.382 and 3.57 μM, respectively. Further in vitro screening of compounds (**52 and 53**) against the amastigote forms of *T. cruzi* showed inhibitory effects with IC$_{50}$ values of 77.0 and 72.0 μM, respectively [72]. Szajnman et al. [73] described the inhibitory effect of a series of 2-alkylaminoethyl-1,1-bisphosphonic acids (compounds (**54–60**)) [1-[(n-But-1-ylamino)ethyl] 1,1-bisphosphonic acid (**29**); 1-[(n-Pent-1-ylamino)ethyl] 1,1-bisphosphonic acid (**30**); 1-[(n-Hex-1-ylamino)ethyl] 1,1-bisphosphonic acid (**31**); 1-[(n-Hept-1-ylamino)ethyl] 1,1-bisphosphonic acid (**32**); 1-[(n-Oct-1-ylamino)ethyl] 1,1-bisphosphonic acid (**54**); 1-[(3-Methyl-but-1-ylamino)ethyl] 1,1-bisphosphonic acid (**33**); and 1-[(tert-Butylamino)ethyl] 1,1-bisphosphonic acid (**55**)] against the intracellular form of *T. cruzi*, exhibiting IC$_{50}$ values of 2.28, 1.84, 0.49, 0.058, 1.014, 0.42 and 1.21 μM, for compounds (**29–32**, **54**, **33,** and **55**), respectively. These compounds were further assayed for inhibitory activity on *T. cruzi* farnesyl diphosphate synthase, and the results showed IC$_{50}$ values of 2.28, 1.84, 0.49, 0.058, 1.014, 0.42, and 1.21 μM for compounds (**29–32**, **54**, **33,** and **55**), respectively [73]. Additionally, 1-[(n-dodec-1-ylamino)ethyl] 1,1-bisphosphonic acid (**56**) inhibited the growth of a more clinically relevant form of *Trypanosoma cruzi* with an IC$_{50}$ value of 0.67 μM compared with benznidazole, the positive control considered (IC$_{50}$: 2.77 μM). This compound was also effective against amastigote forms of *T. cruzi*, exhibiting an IC$_{50}$ value of 0.67 μM. The target enzyme *T. cruzi* farnesyl diphosphate synthase (TcFPPS) was also inhibited by this compound (IC$_{50}$ value: 0.81 μM) [74]. Furthermore, Rodrígues-Poveda et al. [9] synthesized a series of 2-alkyl aminoethyl-1,1-bisphosphonic acids (compounds (**29–32**, **55**, **57,** and **34**)) as potent inhibitors of *T. cruzi* farnesyl pyrophosphate synthase with IC$_{50}$ values of 2.28, 1.84, 0.49, 0.058, 1.21, 69.8, and 57.3 nM, respectively. Further in vitro studies demonstrated that compounds (**29–32**, **55**, **57,** and **34**) inhibit the growth of *T. cruzi* amastigotes with IC$_{50}$ values of 4.8, 0.54, 0.84, 10.0, 10.0, 1.39, and >10 nM, respectively [9].

Additionally, 2-(n-propylamino) ethane-1,1-diyl]bisphosphonic acid (**58**) and [2-(n-heptylamino)ethane-1,1-diyl]bisphosphonic acid (**32**) afforded IC$_{50}$ values of 38.0 and 58.0 nM, respectively, when tested on *T. cruzi* farnesyl diphosphate synthase [75]. In a paper published by Recher et al. [76], compounds (**59–61**) exhibited ED$_{50}$ values of 15.8, 12.8, and 22.4 μM, respectively, when tested against amastigotes of *T. cruzi*. These cellular activities matched the inhibition of the enzymatic activity of the target enzyme (TcFPPS) with IC$_{50}$ values of 6.4, 1.7, and 0.097 μM, respectively [76].

Moreover, Lindert et al. [77] described the inhibitory effect of a series of compounds (**62–71**) on farnesyl diphosphate synthase with IC$_{50}$ values of 1.8, 1.9, 2.5, 7.0, 10.7, 13.7, 20.3, 21.0, 22.3, and 35.0 μM for compounds (**62–71**), respectively [77]. In a paper published by Liu et al. [12], the quinone methide celastrol (**72**) showed an inhibitory effect on *Trypanosoma brucei* farnesyl diphosphate synthase (FPPS) with an IC$_{50}$ value ∼20 μM [12].

In 2014, Aripirala et al. [78] reported the inhibitory potential of two nitrogen-containing bisphosphonates [1-(2-hydroxy-2,2-diphosphonoethyl)-3-phenylpyridinium (**73**) and 3-fluoro-1-(2-hydroxy-2,2-diphosphonoethyl)-pyridinium (**74**) in inhibiting *L. major* farnesyl diphosphate synthase with Ki values of 9 and 50 μM, respectively, vs. zoledronate (11 μM) [78].

Recently, Galaka et al. [79] reported the inhibitory effect of three 1-alkylaminomethyl-1,1-bisphosphonic acids (compounds (**31**, **32,** and **56**)), with $IC_{50}$ values of 0.49, 0.058, and 0.81 μM, respectively, against *Trypanosoma brucei* farnesyl pyrophosphate synthase. After incubation of compounds 31, 32, and 56 with *T. cruzi*, $IC_{50}$ values of 0.84, 10.0, and 0.67 μM were obtained, respectively, vs. benznidazole ($IC_{50}$: 2.58 μM) [79].

As already discussed, farnesyl diphosphate synthase is one of the crucial enzymes involved in the isoprenoid pathway, which is required for sterol biosynthesis. In the present study, forty (40) inhibitors of farnesyl diphosphate synthase were recorded across the literature. Among these compounds, nineteen (19) were reported to be active against *Trypanosoma cruzi* and included compounds (**41**, **43**, **45** and **49**) ($IC_{50}$ values: 21.4, 18.1, 65.8 and 22.36 μM, respectively; amastigote forms of *T. cruzi*, [66]); compounds (**52** and **53**) ($IC_{50}$ values of 77.0 and 72.0 μM, respectively, [72]; compounds (**29–32**, **54**, **33**, and **55**), $IC_{50}$ values: 2.28, 1.84, 0.49, 0.058, 1.014, 0.42 and 1.21 μM, respectively, [73]; 1-[(n-dodec-1-ylamino)ethyl] 1,1-bisphosphonic acid (56); $IC_{50}$: 0.67 μM; clinically relevant form of *T. cruzi*, [74]; compounds (**29–32**, **55**, **57** and **34**) ($IC_{50}$s: 4.8, 0.54, 0.84, 10.0, 10.0, 1.39 and >10 nM, respectively; *T. cruzi* amastigotes, [9]; compounds (**59**, **60,** and **61**) ($ED_{50}$s: 15.8, 12.8, and 22.4 μM, respectively; *T. cruzi* amastigotes, [76]; and compounds (**31**, **32** and **56**) ($IC_{50}$s: 0.84, 10.0 and 0.67 μM, respectively; *T. cruzi*, [79]) (Table 2). This observation clearly verifies the target-based antiparasitic drug discovery approach, in which the activity of a compound can be screened by targeting enzymes that are crucial for the survival of the parasite.

**Table 2.** Potential inhibitors of farnesyl diphosphate synthase.

| Serial Number | Compound Name | Model | Significant Results | Reference |
|---|---|---|---|---|
| 41–50 | Compounds **41–45**, **46**, **47**, **48**, **49** and **50** | *Trypanosoma cruzi* farnesyl pyrophosphate synthase *Trypanosoma brucei* farnesyl pyrophosphate synthase | $IC_{50}$ values: 42.83, 1.94, 2.37, 9.36, 8.45, 5.71, 5.67, 4.54, 19.73 and 4.25 μM for compounds **41–50**, respectively. $IC_{50}$ values: 3.12, 0.66, 3.57 and 4.54 μM, for compounds **41–45**, respectively | [66] |
| 51 | Risedronate (**51**) | NS | $IC_{50}$: 65.4 μM | [71] |
| 52–53 | Alkyl 1-amino-1,1-bisphosphonates (compounds **52** and **53**) | *T. cruzi* farnesyl pyrophosphate synthase | $IC_{50}$ values 0.382 and 3.57 μM, for compounds **10** and **11**, respectively; compounds **52** and **53**, respectively. | [72] |
| 54–55 | [1-[(n-But-1-ylamino)ethyl] 1,1-bisphosphonic acid (**29**); 1-[(n-Pent-1-ylamino)ethyl] 1,1-bisphosphonic acid (**30**); 1-[(n-Hex-1-ylamino)ethyl] 1,1-bisphosphonic acid (**31**); 1-[(n-Hept-1-ylamino)ethyl] 1,1-bisphosphonic acid (**32**); 1-[(n-Oct-1-ylamino)ethyl] 1,1-bisphosphonic acid (**54**); 1-[(3-Methyl-but-1-ylamino)ethyl] 1,1-bisphosphonic acid (**33**); and 1-[(tert-Butylamino)ethyl] 1,1-bisphosphonic acid (**55**)] | *T. cruzi* farnesyl diphosphate synthase | $IC_{50}$s: 2.28, 1.84, 0.49, 0.058, 1.014, 0.42 and 1.21 μM, for compounds (**29–32**, **54**, **33** and **55**), respectively; | [73] |
| 56 | 1-[(n-dodec-1-ylamino)ethyl] 1,1-bisphosphonic acid (**56**) | *T. cruzi* farnesyl diphosphate synthase | $IC_{50}$: 0.67 μM, vs. benznidazole ($IC_{50}$: 2.77 μM) | [74] |
| 57 | 2-alkylaminoethyl-1,1-bisphosphonic acids (compounds (**29–32**, **55**, **57** and **34**)) | *T. cruzi* farnesyl pyrophosphate synthase | $IC_{50}$ values: 4.8, 0.54, 0.84, 10.0, 10.0, 1.39 and >10 nM, for compounds **29–32**, **55**, **57** and **34**, respectively | [9] |
| 58 | 2-(n-propylamino) ethane-1,1-diyl]bisphosphonic acid (**58**) and [2-(n-heptylamino)ethane-1,1-diyl]bisphosphonic acid (**32**) | *T. cruzi* farnesyl diphosphate synthase | $IC_{50}$ values: 38.0 and 58.0 nM, for **58** and **32**, respectively. | [75] |
| 59–61 | Compounds (**59**, **60** and **61**) | Target enzyme TcFPPS | $IC_{50}$ values: 6.4, 1.7, and 0.097 μM, for compounds **59–61**, respectively | [76] |
| 73–74 | [1-(2-hydroxy-2,2-diphosphonoethyl)-3-phenylpyridinium (**73**) and 3-fluoro-1-(2-hydroxy-2,2-diphosphonoethyl)-pyridinium (**74**) | *L. major* farnesyl diphosphate synthase | Ki values: 9 and 50 μM for compounds **73** and **74**, respectively, vs. zoledronate (11 μM) | [78] |

**Table 2.** *Cont.*

| Serial Number | Compound Name | Model | Significant Results | Reference |
|---|---|---|---|---|
| 3 | N4,N4'-bis(3-(4,5-dihydro-1H-imidazol-2-yl)phenyl)-[1,1'-biphenyl]-4,4'-dicarboxamide (**62**); N1,N3-bis(4-(4,5-dihydro-1H-imidazol-2-yl)phenyl)isophthalamide (**63**); 2-amino-N1-(4-(4-methyl-4,5-dihydro-1l4,3l2-imidazol-2-yl)phenyl)-N4-(4-(4-methyl-4,5-dihydro-1H-imidazol-2-yl)phenyl)terephthalamide (**64**); N1,N4-bis(2-(4,5-dihydro-1H-imidazol-2-yl)phenyl)terephthalamide (**65**); N1,N4-bis(3-carbamoyl-5-(4,5-dihydro-1H-imidazol-2-yl)phenyl)terephthalamide (**66**); N1,N4-bis(3-(4,5-dihydro-1H-imidazol-2-yl)phenyl)terephthalamide (**67**); 2-amino-N1,N4-bis(4-(4,5-dihydro-1H-imidazol-2-yl)phenyl)terephthalamide (**68**); 2-amino-N1,N4-bis(3-(4,5-dihydro-1H-imidazol-2-yl)phenyl)terephthalamide (**69**); 2-amino-N1,N3-bis(4-(4,5-dihydro-1H-imidazol-2-yl)phenyl)isophthalamide (**70**); N1,N4-bis(4-(4,5-dihydro-1H-imidazol-2-yl)-3-hydroxyphenyl)terephthalamide (**71**). | Human FPPS | $IC_{50}$: 1.8, 1.9, 2.5, 7.0, 10.7, 13.7, 20.3, 21.0, 22.3 and 35.0 μM, for compounds **62–71**, respectively | [77] |
| 5 | Quinone methide celastrol (**72**) | *Trypanosoma brucei* farnesyl diphosphate synthase (FPPS) | $IC_{50}$ value ~20 μM | [12] |
| 11 | Compounds (**31**, **32** and **56**) | *Trypanosoma brucei* farnesyl pyrophosphate synthase | $IC_{50}$ values: 0.49, 0.058 and 0.81 μM for compounds (**31**, **32** and **56**), respectively | [79] |

NS: Not Specified.

## 9. Critical Assessment and Discussion

The present study aims to emphasize the gaps in our knowledge on the following: (i) the use of squalene synthase and farnesyl diphosphate synthase as potential target proteins for the discovery of new anti-*Trypanosomatidae* treatments; and (ii) the issues thus far not suitably explored.

As already discussed, a total of forty compounds were reported to inhibit squalene synthase. Additionally, forty compounds (40) were found to inhibit the activity of farnesyl diphosphate synthase. Notably, among the inhibitors of squalene synthase, eight compounds (**3**, **22**, **23**, and **29–33**) were found to be active against *Leishmania* spp., whereas five compounds (**22–26**) exhibited anti-*T. cruzi* activity. For the inhibitors of farnesyl diphosphate synthase, nineteen compounds (**41**, **43**, **45,** and **49**; **52** and **53**; **29–32**, **54–57**, **33**, **34**, **59–61**) displayed anti-*Trypanosoma cruzi* activity.

However, the following research gaps were noted: (i) lack of appropriate controls (negative and positive) in experiments; (ii) in most of the papers, the enzymatic assays were not followed by phenotypic screening of the compounds for anti-Trypanosomatid activity; (iii) there was a lack of data regarding the cytotoxicity of bioactive compounds against human mammalian cells; and (iv) noteworthy, data regarding enzymatic assays and phenotypic screening in vivo are scarce.

As already discussed, reverse pharmacology, or target-based drug discovery, has been used to identify lead compounds against the parasites responsible for leishmaniasis and trypanosomiasis [62]. However, new drug discovery strategies, such as the dereplication technique, can be used to identify ubiquitous natural products or even known compounds that are potentially active against the *Trypanosomatidae*. As already discussed, the increase in partnerships between the public and private sectors, and the fulfillment of concerns regarding research funds might sustain anti-*Trypanosomatidae* drug discovery [63]. Although rational design of multitargeting agents is extremely complex in polypharmacology modeling, this concept could be useful in drug discovery as it involves the interaction of drug molecules with multiple targets, which may interfere with a single or multiple disease pathways [80]. In addition to these activities, toxicity studies and the pharmacokinetics of the most promising compounds should be investigated. Phenotypic screening of inhibitors of squalene synthase and farnesyl diphosphate synthase should be performed for anti-trypanosomatid activity against *Leishmania* and *Trypanosoma* species to verify the activity on the parasite. In vitro and in vivo toxicity studies of the bioactive compounds should be carried out to evaluate their selectivity.

## 10. Authors' Opinion on the Topic

Neglected tropical diseases, such as leishmaniases and trypanosomiases, are prevalent in several sub-Saharan African and South American countries [2]. These diseases are caused by trypanosomatid parasites that interact with a wide range of insects and mammals to complete their life cycles. In general, *Trypanosoma* and *Leishmania* species are the parasites causing trypanosomiases and leishmaniases, respectively. A number of approaches, including phenotypic drug discovery and target-based drug discovery, have been used to identify anti-*Trypanosomatidae* compounds. We have summarized and discussed inhibitors of squalene synthase and farnesyl diphosphate synthase in this manuscript. Notably, a number of inhibitors of these enzymes exhibited moderate to high anti-trypanosomatid activity upon in vitro phenotypic screening, attesting to the involvement of squalene synthase and farnesyl diphosphate synthase in the pathogenesis of trypanosomatids. Indeed, squalene synthase and farnesyl diphosphate synthase are the main enzymes that intervene in the isoprenoid pathway and the biosynthesis of ergosterol and other sterol compounds in trypanosomatids. The five best inhibitors of squalene synthase that displayed superior anti-trypanosomatid activity included compounds (**30**, **31,** and **33**) (IC$_{50}$ values: 0.54, 0.84, and 0.94 nM, respectively; [9]) and compounds (**22** and **23**) (IC$_{50}$s: 1.7 and 14.7 nM, respectively; [59]). The five best inhibitors of farnesyl diphosphate synthase that exhibited higher anti-trypanosomatid activity included compound (**56**); IC$_{50}$: 0.67 µM [74]; (**30** and

**31**) (IC$_{50}$s: 0.54 and 0.84 nM, respectively [9]; and (**31** and **56**) (IC$_{50}$s: 0.84 and 0.67 μM, respectively) [79]. More importantly, three compounds that showed anti-*Trypanosomatidae* activity were identified as inhibitors of both farnesyl diphosphate synthase and squalene synthase and included compounds (**30**, **31,** and **33**). Based on these observations, squalene synthase and farnesyl diphosphate synthase are potential targets for the identification of hit compounds that can serve as scaffolds for the discovery of anti-Trypanosomatid drugs. However, more studies are needed to potentially verify this target-based drug discovery against traditional phenotypic drug screening, as almost half of the reported squalene synthase and farnesyl diphosphate synthase inhibitors have not been tested on trypanosomatids. Thus, doors are opened for researchers who work on anti-trypanosomatid drug discovery to screen such inhibitors. More in vitro and in vivo toxicity studies, pharmacokinetics, and structural medications of the most prominent scaffolds are desired to improve and select the best potential hit candidates that can be pursued further for anti-trypanosomatid drug discovery. Although several strategies have been established for anti-*Trypanosomatidae* drug discovery, (i) testing for inhibitors against disease-relevant targets, (ii) screening for growth inhibitors against the whole parasite through in vitro cell culturing, and (iii) searching for anti-trypanosomatid therapies among previously known drugs (repositioning of known molecules) are mainly employed to identify hit compounds that can serve as scaffolds for the development of new agents against trypanosomiasis and leishmaniasis [81,82].

## 11. Conclusions and Future Perspectives

Addressing tropical diseases, such as leishmaniasis and trypanosomiases, requires cross-sectoral approaches for drug discovery that span from phenotypic-based screening to target-based screening. However, information pertaining to the molecular pathogenesis of trypanosomatids is still controversial. In-depth understanding of the pathogenic mechanisms of action of *Leishmania* and *Trypanosoma* parasites might unravel difficulties in discovering new vaccines and treatments against leishmaniasis and trypanosomiasis. In fact, two key enzymes that are involved in the biosynthesis of cholesterol in humans and ergosterol in trypanosomatids include farnesyl diphosphate synthase and squalene synthase. In fact, these enzymes are involved in the isoprenoid pathway and sterol biosynthesis, respectively, to generate cholesterol in humans and ergosterol in trypanosomatids. Notably, the latter compound (ergosterol) is also essential for parasite survival. Thus, inhibitors of these enzymes (farnesyl diphosphate synthase and squalene synthase), which contribute to ergosterol biosynthesis, can be used as potential targets for the development of anti-trypanosomatid drugs.

In fact, the present study aimed to summarize recent developments in anti-*Trypanosomatidae* hit compounds using a target-based approach using farnesyl diphosphate synthase and squalene synthase as the target enzymes. Validation of the anti-trypanosomatid activity of farnesyl diphosphate synthase and squalene synthase inhibitors using phenotypic screening against trypanosomatids is also highlighted. Thus, there is no denying that inhibitors of farnesyl diphosphate and squalene synthases can afford potentially active compounds against *Leishmania* and *Trypanosoma* species. However, more in vitro and in vivo phenotypic screenings are needed to verify the potential activity of these inhibitors against *Leishmania* and *Trypanosoma* species. More toxicity studies and pharmacokinetics and structural modifications are recommended to identify the most active inhibitors as starting points for the discovery of new anti-trypanosomatid drugs.

**Author Contributions:** Conceptualization, B.P.K. and F.F.B.; methodology, B.P.K. and F.F.B.; software, B.P.K.; validation, B.P.K. and F.F.B.; formal analysis, B.P.K.; investigation, B.P.K.; resources, B.P.K.; data curation, B.P.K.; writing—original draft preparation, B.P.K.; writing—review and editing, B.P.K. and F.F.B.; visualization, B.P.K.; supervision, F.F.B.; project administration, F.F.B.; funding acquisition, B.P.K. and F.F.B. All authors have read and agreed to the published version of the manuscript.

**Funding:** This research received no external funding.

**Institutional Review Board Statement:** Not applicable.

**Informed Consent Statement:** Not applicable.

**Data Availability Statement:** Not applicable.

**Conflicts of Interest:** The authors declare no conflict of interest.

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
