# Peer review of "Inhibitors of Farnesyl Diphosphate Synthase and Squalene Synthase: Potential Source for Anti-Trypanosomatidae Drug Discovery"

_ddc, doi:10.3390/ddc2030032_

Round 1
Reviewer 1 Report
The review entitled “Inhibitors of farnesyl diphosphate synthase and squalene synthase: Potential source for anti-Trypanosomatidae drug discovery” is a fascinating literature study regarding the potential drugs used to treat leishmaniasis, sleeping sickness, and Chagas disease, diseases in which there is resistance to current medication, as in the case of antibiotics.
The work is well done, interesting and useful, both from the point of view of new chemical structures and biological potential.
I think that some minor changes are needed:
· The abbreviations must be explained the first time they are used in the text.
Ex. line 71 “for VL and oral miltefosine for CL, ML” the explanations are given in lines 219-220
· Line 87 24-ethyl-cholesta-5,7,22-trien-3 beta ol should be written 24-ethyl-cholesta-5,7,22-trien-3-beta-ol
· a figure with the chemical structures of the most used anti-Trypanosomatidae drugs should be added
· for words of Latin origin (in vitro (line 320), de novo (line 335)), italics must be used
· dehydro quinuclidine should be written dehydro-quinuclidine (lines 426, 427)
· for chemical names do not use spaces (line 443, 617)
· lines 457-459 why the authors preferred the 29-33 and 34 expressions instead of 29-34?
· I think it would have been more useful if figures 3 and 4 were expanded into smaller figures, presented where the compounds are described in the text
Author Response
Response to the reviewer 1 comments point by point
Manuscript ID: ddc-2483981
Title: Inhibitors of farnesyl diphosphate synthase and squalene synthase:
Potential source for anti-Trypanosomatidae drug discovery.
Dear Reviewer,
We have revised the manuscript in accordance to your comments and suggestions. The point wise changes are as follows:
Reviewer 1
The review entitled “Inhibitors of farnesyl diphosphate synthase and squalene synthase: Potential source for anti-Trypanosomatidae drug discovery” is a fascinating literature study regarding the potential drugs used to treat leishmaniasis, sleeping sickness, and Chagas disease, diseases in which there is resistance to current medication, as in the case of antibiotics.
The work is well done, interesting and useful, both from the point of view of new chemical structures and biological potential.
Reply: We thank the reviewer for its optimistic view toward our manuscript.
I think that some minor changes are needed:
- The abbreviations must be explained the first time they are used in the text.
Ex. line 71 “for VL and oral miltefosine for CL, ML” the explanations are given in lines 219-220
Reply: We agree with the reviewer’s inquisitiveness. The full forms for VL, CL and ML have now been incorporated in the appropriate lines.
- Line 87 24-ethyl-cholesta-5,7,22-trien-3 beta ol should be written 24-ethyl-cholesta-5,7,22-trien-3-beta-ol
Reply: As recommended, “87 24-ethyl-cholesta-5,7,22-trien-3 beta ol” has now been replaced by “24-ethyl-cholesta-5,7,22-trien-3-beta-ol”
- a figure with the chemical structures of the most used anti-Trypanosomatidaedrugs should be added
Reply: A figure with the chemical structures of the most used anti-Trypanosomatidae drugs has now been added.
- for words of Latin origin (in vitro (line 320), de novo (line 335)), italics must be used
Reply: Words of latin origin, such as “in vitro” and “de novo” have now been written in italics.
- dehydro quinuclidine should be written dehydro-quinuclidine (lines 426, 427)
Reply: “dehydro quinuclidine” has now been replaced by “dehydro-quinuclidine”.
- for chemical names do not use spaces (line 443, 617)
Reply: We agree with the reviewer’s suggestion. The spaces have now been omitted.
- lines 457-459 why the authors preferred the 29-33 and 34 expressions instead of 29-34?
Reply: “29-33 and 34” has now been replaced by “29-34”
- I think it would have been more useful if figures 3 and 4 were expanded into smaller figures, presented where the compounds are described in the text
Reply: Figure 3 and 4 have now been expanded into smaller figures as recommended.
We request the reviewer for re-evaluation and kind consideration.

Reviewer 2 Report
The manuscript presents an interesting and competent review of the potential inhibitors of squalene synthase and farnesyl phosphate synthase with anti-Trypanosomatidae activity. In view of the gravity of the problem of neglected tropical diseases, it is a potentially very useful review.
The enzymes of sterol metabolism may be important targets of anti-trypanosomid therapy. However, as discussed by the authors, studies of enzyme inhibition must be followed by an examination of the effects of inhibitors on mammalian cells, and their side effects on the hosts. Moreover, some of these compounds may be substrates for drug efflux pumps of the parasites, which may limit their efficiency.
Line 80: please change “s terol” to “sterol”
Line 87: “24-ethyl-cholesta-5,7,22-trien-3 beta ol”, please change to “24-ethyl-cholesta-5,7,22-trien-3-beta-ol”
Figure 2: The role of zaragozic acids as inhibitors of squalene synthase should be somehow indicated in the scheme (in the present form, their role is unclear in the scheme).
Figure 2: please change “demetylase” to “demethylase”
Author Response
Response to the reviewer 2’s comments point by point
Manuscript ID: ddc-2483981
Title: Inhibitors of farnesyl diphosphate synthase and squalene synthase:
Potential source for anti-Trypanosomatidae drug discovery.
Dear Reviewer,
We have revised the manuscript in accordance to your comments and recommendations. The point wise changes are as follows:
Reviewer 2
The manuscript presents an interesting and competent review of the potential inhibitors of squalene synthase and farnesyl phosphate synthase with anti-Trypanosomatidae activity. In view of the gravity of the problem of neglected tropical diseases, it is a potentially very useful review.
The enzymes of sterol metabolism may be important targets of anti-trypanosomid therapy. However, as discussed by the authors, studies of enzyme inhibition must be followed by an examination of the effects of inhibitors on mammalian cells, and their side effects on the hosts. Moreover, some of these compounds may be substrates for drug efflux pumps of the parasites, which may limit their efficiency.
Reply: We are very thankful for the optimistic view of the reviewer toward our manuscript.
Line 80: please change “s terol” to “sterol”
Reply: We agree with the reviewer’s suggestion. “s terol” has now been replaced by “sterol”
Line 87: “24-ethyl-cholesta-5,7,22-trien-3 beta ol”, please change to “24-ethyl-cholesta-5,7,22-trien-3-beta-ol”
Reply: As recommended, “87 24-ethyl-cholesta-5,7,22-trien-3 beta ol” has now been replaced by “24-ethyl-cholesta-5,7,22-trien-3-beta-ol”
Figure 2: The role of zaragozic acids as inhibitors of squalene synthase should be somehow indicated in the scheme (in the present form, their role is unclear in the scheme).
Reply: We agree with the reviewer’s inquisitiveness. Figure 2 has now been revised as recommended.
Figure 2: please change “demetylase” to “demethylase”
Reply: We agree with the reviewer’s suggestion. In figure 2, “demetylase” has now been replaced by “demethylase”
We request the reviewer for re-evaluation and kind consideration.

Reviewer 3 Report
The review is interesting and valuable, the subject is important and the Authors made a lot of effort to collect all the data. The manuscript is summarized by relevant conclusions and Authors’ opinion.
I have only several suggestions of improvements.
Line 43. “pathological conditions” – I am not sure if it is an appropriate terms, since the accepted definition of pathological conditions is “Abnormal anatomical or physiological conditions and objective or subjective manifestations of disease, not classified as disease or syndrome”. Maybe it would be better to avoid this debatable statement.
Line 87. „24-ethyl-cholesta-5,7,22-trien-3 beta ol” ahould be written as 24-ethyl-cholesta-5,7,22-trien-3β-ol (with Greek letter beta and hyphens), please correct it.
Line 259. “The basic structure of sterol molecules (such as cholesterol, ergosterol, etc.) is derived from acetyl-CoA through two successive pathways, including the mevalonate and isoprenoid pathways.” This sentence can be misleading for some readers, who are rather convinved that “isoprenoids are biosynthesized from isoprenyl diphosphate units, generated by two distinctive biosynthetic pathways:mevalonate pathway and methylerthritol 4-phosphate pathway. “ In this manuscript, the Authors are not mentioned MEP pathway (existing for example in Plasmodium spp.), therefore, speaking about “two pathways” – even if “successive” (and not alternative, and in the case of MVA and MEP) can be wrongly understood. In fact, for the majority of researchers, “isoprenoid pathway” starts from the precursors either of MVA or MEP, so mevalonic pathway is just a part of isoprenoid pathway.
Please avoid this problem, writing simply: The basic structure of sterol molecules (such as cholesterol, ergosterol, etc.) is derived from acetyl-CoA through mevalonate pathway.
Author Response
Response to the reviewer3’s comments point by point
Manuscript ID: ddc-2483981
Title: Inhibitors of farnesyl diphosphate synthase and squalene synthase:
Potential source for anti-Trypanosomatidae drug discovery.
Dear Reviewer,
We have revised the manuscript in accordance to your comments and recommendations. The point wise changes are as follows:
Reviewer 3
The review is interesting and valuable, the subject is important and the Authors made a lot of effort to collect all the data. The manuscript is summarized by relevant conclusions and Authors’ opinion.
Reply: We thank the reviewer for its optimistic view towards our manuscript.
I have only several suggestions of improvements.
Line 43. “pathological conditions” – I am not sure if it is an appropriate terms, since the accepted definition of pathological conditions is “Abnormal anatomical or physiological conditions and objective or subjective manifestations of disease, not classified as disease or syndrome”. Maybe it would be better to avoid this debatable statement.
Reply: We agree with the reviewer’s inquisitiveness. The statement has been omitted as recommended.
Line 87. „24-ethyl-cholesta-5,7,22-trien-3 beta ol” ahould be written as 24-ethyl-cholesta-5,7,22-trien-3β-ol (with Greek letter beta and hyphens), please correct it.
Reply: As recommended, “87 24-ethyl-cholesta-5,7,22-trien-3 beta ol” has now been replaced by “24-ethyl-cholesta-5,7,22-trien-3β-ol”
Line 259. “The basic structure of sterol molecules (such as cholesterol, ergosterol, etc.) is derived from acetyl-CoA through two successive pathways, including the mevalonate and isoprenoid pathways.” This sentence can be misleading for some readers, who are rather convinved that “isoprenoids are biosynthesized from isoprenyl diphosphate units, generated by two distinctive biosynthetic pathways:mevalonate pathway and methylerthritol 4-phosphate pathway. “ In this manuscript, the Authors are not mentioned MEP pathway (existing for example in Plasmodium spp.), therefore, speaking about “two pathways” – even if “successive” (and not alternative, and in the case of MVA and MEP) can be wrongly understood. In fact, for the majority of researchers, “isoprenoid pathway” starts from the precursors either of MVA or MEP, so mevalonic pathway is just a part of isoprenoid pathway.
Please avoid this problem, writing simply: The basic structure of sterol molecules (such as cholesterol, ergosterol, etc.) is derived from acetyl-CoA through mevalonate pathway.
Reply: We agree with the reviewer’s suggestion. The sentence has been amended as recommended.
We request the reviewer for re-evaluation and kind consideration.
